# Towards an understanding of surface effects: Testing of various materials in a small volume measurement chamber and its relevance for atmospheric trace gas analysis

Ece Satar[1,2], Peter Nyfeler[1,2], Céline Pascale[3], Bernhard Niederhauser[3], and Markus Leuenberger[1,2]

[1]Climate and Environmental Physics, Physics Institute, University of Bern, Bern, Switzerland
[2]Oeschger Centre for Climate Change Research, University of Bern, Bern, Switzerland
[3]Federal Institute of Metrology METAS, Bern, Switzerland

**Correspondence:** satar@climate.unibe.ch

**Abstract.** A critical issue for the long-term monitoring of atmospheric trace gases is precision and accuracy of the measurement systems employed. Both measuring and preparing reference gas mixtures for trace gases are challenging due to e.g. adsorption/desorption of the substances of interest on surfaces; this is particularly critical at low amount fraction and/or for reactive gases. Therefore, to ensure the best preparation and measurement conditions for trace gases in very low amount fractions, usage of coated materials is in demand in gas metrology and atmospheric measurement communities. This study focuses on testing potential adsorption/desorption effects for different materials or coatings that are currently used, or may be relevant in future for the measurements of greenhouse gases. For this study we used small volume chambers designed to be used for adsorption studies. Various materials with or without coatings were loaded into the small cylinder to test their adsorption/desorption behavior. We used the aluminum cylinder as the measurement chamber, and glass, aluminum, copper, brass, steel and three different commercially available coatings as test materials. Inserting the test materials into the measurement chamber doubles the available geometric area for the surface processes. The presented experiments were designed to investigate the pressure dependency of adsorption up to 15 bar, and its temperature dependency up to 80 °C for the test materials placed in the measurement chamber. Here, we focused on the species $CO_2$, $CH_4$, CO and $H_2O$ measured by a cavity ring down spectroscopy analyzer. Our results show that the materials currently used in atmospheric measurements are well-suited. The investigated coatings were not superior to untreated aluminum or to stainless steel at the tested pressure ranges, whereas under changing temperature aluminum showed better performance for $CO_2$ (< 0.05 µmol mol$^{-1}$ change in measured amount fractions) than stainless steel (> 0.1 µmol mol$^{-1}$). To our knowledge, this study is one of the first attempts to investigate surface effects of different materials in such a setup for the above-mentioned gases.

## 1 Introduction

Long term atmospheric monitoring of trace gases requires great attention to precision and accuracy. In order to achieve a high level of compatibility for data obtained at different sites and/or at different time, the World Meteorological Organization (WMO) has recommended compatibility goals for measurements of trace gases within its Global Atmosphere Watch (GAW)

Programme (2016). These challenging limits can be achieved not only by regular calibration with standard gases of known composition, but also by limiting any cause of amount fraction alteration. During their relatively long lifetime, in the order of decades, standard gas cylinders may not be stable due to diffusion, leakage, regulator effects, gravimetric fractionation and surface processes (Keeling et al., 2007; Langenfelds et al., 2005). The latter, which encompass adsorption/desorption, are also dependent on temperature, pressure and surface properties. Currently there exists only limited data and a few attempts to quantify these surface processes for $CO_2$ and $CH_4$ (Leuenberger et al., 2015; Miller et al., 2015; Brewer et al., 2018; Schibig et al., 2018). These studies use Langmuir (1918) adsorption theory (Leuenberger et al., 2015; Schibig et al., 2018) and Rayleigh fractionation (Schibig et al., 2018) to explain the enrichment in the amount fractions towards the end of the cylinder lifetime with respect to different flow rates.

Key results of the above-mentioned studies point out that the adsorption behavior is pressure- and temperature- dependent. All mentioned studies used larger volume (10, 29.5 or 50 L) cylinders, which were already in use as standard cylinders. Their approach on filling varied from compressing natural air (Schibig et al., 2018) to gravimetric preparation in synthetic air or in nitrogen (Brewer et al., 2018). In their study, neither Brewer et al. (2018) nor Schibig et al. (2018) observed that their passivation treatment for the aluminum cylinder decreased the surface interaction of $CO_2$.

Langmuir (1918) defines adsorption as the time lag between the condensation of a molecule and its evaporation from a surface. The simplest relation which can be linked to adsorption is the pressure dependency. At higher pressures, the gas molecules are pressed to the cylinder walls, where they are adsorbed to the cylinder surface. As the pressure decreases during the lifetime of a cylinder, these molecules are desorbed from the surface and lead to an enhancement in the amount fraction of the gas. Changes in temperature also affect the equilibrium amount fraction of the adsorbed molecules by varying temperature dependent rate constants of adsorption and desorption.

In this study, we aim at distinguishing these effects among various materials under controlled conditions in a previously characterized measurement chamber (Satar et al., 2019). We limited ourselves to a selection of materials ranging from materials frequently used in atmospheric measurement community to commercially available coatings. Aluminum cylinders are now the state of the art for the measurements of greenhouse gases such as $CO_2$ and $CH_4$ (WMO, 2018). Although not recommended anymore for above-mentioned species, some steel cylinders may still be in use. Additionally, stainless steel pieces are very commonly used as tubings and in pressure regulators, and have contact with the measured gases. Some regulators are made of brass (WMO, 2018), and copper is commonly used as seals in vacuum applications of atmospheric trace gas measurements (Behrens et al., 2008). Moreover, commercially available coatings are increasingly interesting for both atmospheric measurement and metrology communities, since with the improvement of experimental techniques, the demand for higher precision and accuracy in trace gas analysis is growing.

The affinity of adsorption/desorption deviates largely for different species on various surfaces. Some coatings may provide inert, corrosion resistant, or hydrophobic surfaces, and enable usage of metals instead of polymers with ambiguous outgassing effects. According to the current literature, surface losses are critical especially for more reactive gases during the preparation of the standards. In gas metrology community, this issue has already been investigated in more detail i.e. for species such as ammonia using test tubes with various coatings (Vaittinen et al., 2014), for propane and benzene (Lee et al., 2017), and for

monoterpenes in cylinders (Allen et al., 2018). In their study, Vaittinen et al. (2014) observed that some of the commercial coatings reduced the adsorption loss on the stainless steel surface by a factor of ten or more. The atmospheric measurement community makes use of inert coatings of chemically protective barrier of amorphous silicon (Silcotek Corporation) in air core applications (Karion et al., 2010; Andersen et al., 2018), where the surface to volume ratio is large. Diamond-like carbon coatings are not yet commonly used in trace gas analysis, but have found their place in many applications in a range varying from wear and corrosion protection of magnetic storage media to biological implantations (Grill, 1999).

This study contributes to the limited literature on the discussion of surface effects of different materials for the species $CO_2$, $CH_4$, CO and $H_2O$. It is one of the first attempts to systematically investigate the differences among various materials in a reproducible way using a relatively small custom-made aluminum measurement chamber requiring less gas and time for the measurements. In this study, we briefly introduce the setup and the established procedure for the measurements. Then, we proceed with eight material loadings to the measurement chamber, and test their response to pressure and temperature variations.

## 2 Data and Methods

### 2.1 Measurement setup and used materials

In order to understand adsorption/desorption behavior of various materials, high pressure (up to 130 bar) and small volume (5 L) cylinders of aluminum and steel were designed. These cylinders served as measurement chambers in which various test materials can be inserted. Since the aluminum cylinder showed smaller effects with respect to surface effects in the previous study (Satar et al., 2019), we have chosen to use the aluminum cylinder only for the material experiments in order to minimize the background effect related to the measurement chamber. More information and details on the filling history of the cylinders were previously explained (Satar et al., 2019). Here we provide a brief summary: The aluminum cylinder is made of the aluminum alloy AlMg1SiCu (EN AW-6061), and its composition is specifically chosen that it corresponds to the aluminum commonly used in the atmospheric measurement community. This custom-made cylinder consists of three pieces: a body part in the middle with two caps on the sides (Fig. 1a). These pieces are joined by twelve necked-down bolts on each side, and Inconel X750 seals with silver coating are placed in the caps. It is equipped with four stainless steel bellows sealed valves (SS-4H from Swagelok), where the wetted surfaces are solely of stainless steel and do not include any polymers. The connections are from stainless steel and all tubings are of electropolished stainless steel 1/4". At the outlet, the cylinders are equipped with dual stage pressure regulators made of a stainless steel body with a polychlorotrifluoroethylene (PCTFE) seat (64-3441KA412 from Tescom). Pressure transducers are used at low (PTU-S-AC6-31AC from Swagelok), and high (PTU-S-AC160-31AC from Swagelok) pressure sides of the pressure regulators. Temperature sensors spanning a range from –35 °C to +100 °C (AF25.PT100 from Thermokon) are placed on the outer cylinder surfaces. All measured temperature and pressure data were read and logged by a signal converter (midi logger GL820 from Graphtec). On the measurement line between the pressure regulator and the Picarro Cavity-Ring Down Spectroscopy analyzer (CRDS) G2401 either an electropolished stainless steel

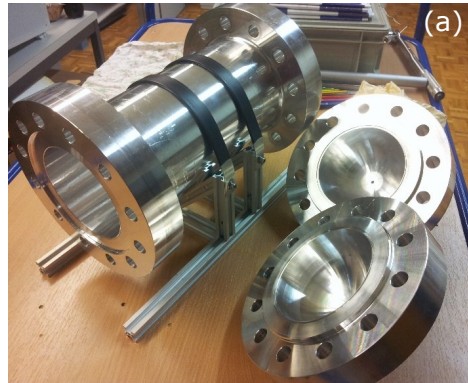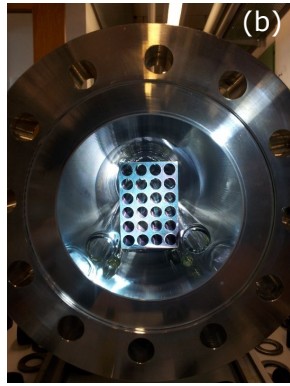

**Figure 1. (a)** Custom made cylinder of three pieces. **(b)** Material loading into the cylinder. The metal blocks are placed on the glass ladder, and two rod shaped glass pieces support the material from the sides.

1/4" tubing, a mass flow controller (358 Series from ANALYT-MTC) or a multiport valve (EMT2CSD6MWE from VICI AG) was placed.

The fillings were done using compressed air from high pressure 50 L aluminum cylinders (LUX3586 and LUX3575). These two cylinders are called the mother cylinders and their air content the mother mixture from here on. A mother cylinder was directly connected to a small expansion volume (0.5 L) made of stainless steel (316L-HDF4-500 from Swagelok). In addition to the mother mixture, another mixture of comparable content and from a cylinder of comparable material and equipment to the mother cylinder was measured to check the stability of the measurement device. This mixture (from cylinder LUX3579) is referred to as the working gas. All three cylinders were filled by Carbagas, Switzerland with compressed air according to their own protocol. The filling history of the cylinders is known only to the extent that the cylinders were filled with compressed air only. In order to test for higher amount fractions of CO, the mother mixtures were spiked: a known amount of pure CO gas was injected into a known volume (60 mL), and was pushed into the sample cylinder using another compressed air mixture as carrier gas. For example, after spiking the mother mixture, the composition of LUX3575 was 428.59 µmol mol$^{-1}$, 1083.73 nmol mol$^{-1}$, 2132.93 nmol mol$^{-1}$, <15 µmol mol$^{-1}$ for $CO_2$, CO, $CH_4$ and $H_2O$.

Material loadings into the cylinder were conducted as follows: glass pieces were inserted in order to avoid sharp metal-metal contact points between the sample pieces and the cylinder inner surface. These consisted of a ladder and two rod shaped glass pieces (Fig. 1b). Then, on top of the ladder-shaped glass piece, two metal blocks were placed. Each metal block has the dimensions 100x74x50 mm, and has 24 drill holes of 1 cm in diameter. The two blocks have in total comparable (factor of 1.17) geometric area to the cylinder inner surface. Aluminum (AlMgSi1), copper (CuETP), brass (CuZn39Pb3), steel (316L), and three different commercially available coatings on steel (SilcoNert®2000, Dursan®, and CERODEM® diamond-like carbon (DLC)) were used as test materials. Glass pieces and metal blocks without coatings underwent a cleaning procedure consisting of ultrasonic bath with a diluted solution of a mildly alkaline commercial cleaning agent (Deconex HT1201), and vacuum oven drying at 120 °C.

Since the cylinder was exposed to outside air in between loadings of different materials, a specific cleaning procedure was applied to eliminate water vapor. The measurement chamber was first pumped down to 0.05 mbar using a dry piston vacuum pump (EcoDry M15 from Leybold), and then filled with 2 bar of $N_2$, and pumped again while heating with a heat gun. After three fill-pump-heat cycles of 30 minutes each, the cylinder was filled with $N_2$, and left for cooling overnight. During these heating cycles, the surface temperature of the sample cylinder increased up to 60 °C. The following morning, the cylinder was pumped down to 0.05 mbar, and filled with compressed air through expansion. The desired pressure in the small cylinder was achieved by repeating the expansion step several times. An hour was allowed for equilibration prior to starting the measurements.

## 2.2 Measurement sequence and data collection

Figure 2 shows a scheme of the measurement setup. For each material loading, temperature and pressure experiments were conducted using the same procedure as before (Satar et al., 2019). The experiments were conducted using a Picarro G2401 CRDS analyzer enabling measurements of $CO_2$, CO, $CH_4$ and $H_2O$. According to the specification sheet of the analyzer, 5-minute, 1-$\sigma$ standard deviation is <0.2 µmol mol$^{-1}$, <1.5 nmol mol$^{-1}$, <0.5 nmol mol$^{-1}$, and <50 µmol mol$^{-1}$ for the species $CO_2$, CO, $CH_4$ and $H_2O$, respectively. In order to investigate the material's pressure dependency, the cylinder was filled through expansion from the mother cylinder to around 15 bar, and was evacuated through the Picarro analyzer. Each sample material loading had at least three replicates for both temperature and pressure runs with the exception of the blank cylinder (Table 1). Bracketing each measurement, the working gas cylinder was measured to check the stability of the measurement device. The measurement sequence for an individual loading was established in the following order: The first two runs were pressure experiments. These were followed by three cycles of temperature experiments. Lastly, the cylinder underwent the third pressure experiment. This order enables the detection of any changes in pressure response after heating cycles. Table 1 shows an overview of the experiments presented in this study.

For the pressure dependency experiments, data analysis was based on Satar et al. (2019). There was no flow regulation after the pressure regulator prior to the analyzer inlet. At the beginning of the experiment the flow rate was 220 mL min$^{-1}$ (STP) and towards the end of the experiment it decreased to 15 mL min$^{-1}$ (STP). The end point of the experiments was set to a fixed internal parameter of the measurement device called the "outlet valve". This value can easily be linked to the outflow from the instrument, which corresponds to about 15 mL min$^{-1}$ at STP. For better comparability among the measurements, measured amount fractions were subtracted from the mean of the first hour of measurements for each run. In order to eliminate instrumental noise, 5-minute means of these differences were calculated. In this study, all reported values are in amount fraction differences ($\Delta CO_2$, $\Delta CH_4$, $\Delta CO$, $\Delta H_2O$).

In order to investigate the temperature dependency, the cylinder was placed into a climate cabinet (ACS Challenge 600) at the Swiss Federal Institute of Metrology (METAS). The temperature of the cabinet was set to –10 °C , 20 °C, 50 °C and 80 °C , with 30 °C increments, heated or cooled within an hour (Fig. 3). The temperature was kept constant for four hours at each level, of which during the last 35 minutes the material loaded cylinder was measured. These measurements were bracketed by working gas measurements which had not experienced any temperature changes. A multiport valve was used to switch between

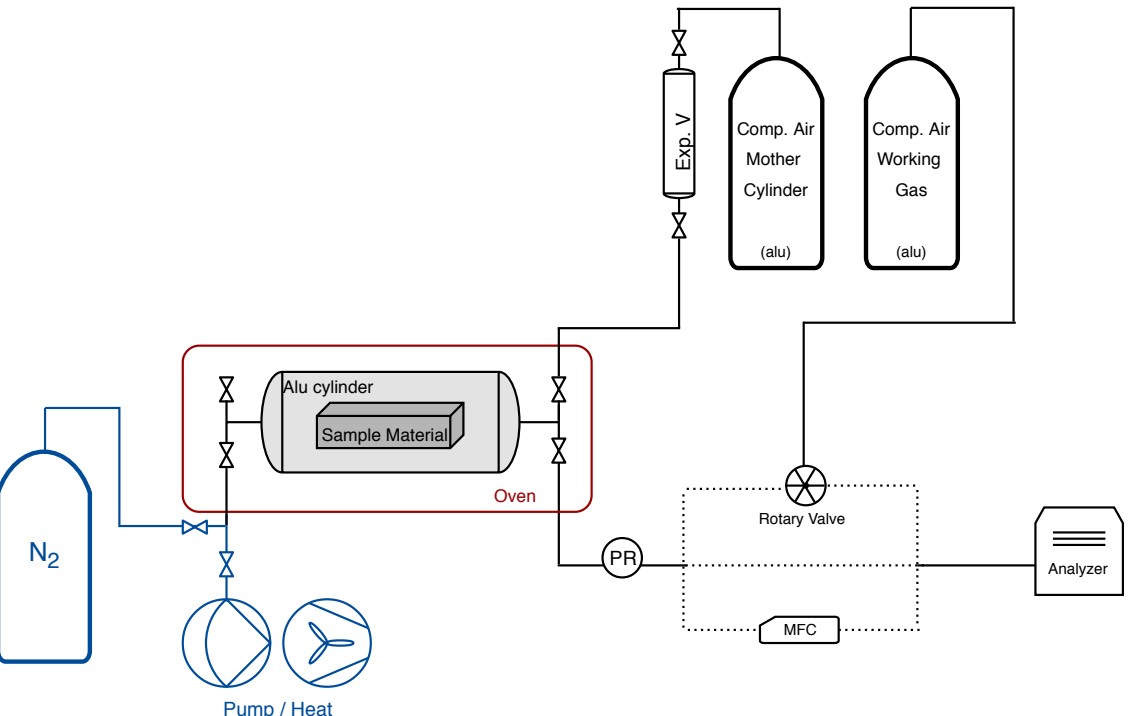

**Figure 2.** Schematic of the experimental setup. The aluminum cylinder is placed in the oven (denoted by the red box). The cylinder is filled through the expansion volume from the mother cylinder. At the outlet of the cylinder, the dashed lines show the three possible paths into the analyzer: through the rotary valve, direct tubing or mass flow controller (MFC). The equipment related to the cleaning procedure is denoted in blue.

the small cylinder and the working gas. A full temperature cycle lasted 34 hours. For the data analysis, for each temperature step the first 10 minutes of the measurements were not included in order to allow time for equilibration; the mean of the remaining 25 minutes was calculated. Then, differences for each temperature level were calculated with respect to the measurements at
5   20 °C.

## 3   Results

### 3.1   Pressure experiments

In Fig. 4, an overview of all measured species is shown. For each run, we calculated the amount fraction differences with respect to the initial amount fraction and selected the maximal difference. Maximal difference was chosen to highlight the maximum
10   possible effect related to desorption. Since all data was processed using the same criterion, the effects originating from the

**Table 1.** An overview of data included in this study. The pressure values indicate the pressure in the small cylinder at the beginning of each replicate run.

| Experiment | Type | Pressure [bar] | Number of replicates | Mother cylinder |
|---|---|---|---|---|
| Blank | Pressure | 13.9; 13.9 | 2 | LUX3575 |
| Blank | Temperature | 15.8; 15.0; 16.7 | 3 | LUX3586 |
| Glass | Pressure | 13.3; 13.0 | 2 | LUX3586 |
| Glass | Temperature | 15.3; 14.7; 14.5 | 3 | LUX3586 |
| Aluminum | Pressure | 13.6; 13.0; 16.3 | 3 | LUX3575 |
| Aluminum | Temperature | 15.3; 14.7; 14.7 | 3 | LUX3586 |
| Steel | Pressure | 16.0; 15.4 | 2 | LUX3586 |
| Steel with VICI | Pressure | 16.4 - 20.3 | 7 | LUX3586 |
| Steel MFC | Pressure | 15.5; 15.0 | 2 | LUX3586 |
| Steel | Temperature | 12.7; 18.5; 18.3 | 3 | LUX3586 |
| SilcoNert®2000 | Pressure | 14.5; 13.7; 14.0 | 3 | LUX3586 |
| SilcoNert®2000 | Temperature | 13.9; 14.0; 16.1 | 3 | LUX3586 |
| Glass | Pressure | 17.1; 16.7; 16.0 | 3 | LUX3575 |
| Glass | Temperature | 16.8; 16.9; 16.7 | 3 | LUX3575 |
| Dursan® | Pressure | 16.3; 9.2; 15.5; 12.3 | 4 | LUX3575 |
| Dursan® | Temperature | 15.8; 15.5; 15.0 | 3 | LUX3575 |
| DLC | Pressure | 13.0; 13.5; 18.3 | 3 | LUX3575 |
| DLC | Temperature | 13.9; 19.6; 19.5 | 3 | LUX3575 |
| Copper | Pressure | 15.4; 14.9; 13.4 | 3 | LUX3575 |
| Copper | Temperature | 15.5; 14.7; 14.6 | 3 | LUX3575 |
| Brass | Pressure | 18.1; 17.3; 15.4 | 3 | LUX3575 |
| Brass | Temperature | 18.4; 17.3; 16.9 | 3 | LUX3575 |

analyzer is cancelled out and we focus on the differences between the materials. We grouped replicate runs of each material loading together and showed the calculated maximal differences in the boxplots. The median is denoted by the horizontal line, whereas the mean is shown by the square. Since for most cases only 3 replicates are present, the $1^{st}$ quartile is the mean of the minimum and the median, whereas the $3^{rd}$ quartile is the mean of the median and the maximum value. For clarity, data points used for the box plots are also shown and they are denoted by the black points. In Fig. 4a and b, $CO_2$ amount fractions are plotted: the first panel includes all materials, whereas the second is a zoom-in aiming to distinguish smaller differences among the material loadings. For $CO_2$, we were able to detect significant changes only for Dursan®, where the final amount fraction was about 10 times higher than all other materials. For CO and $CH_4$, the maximum difference in the amount fractions did not exceed 6 nmol mol$^{-1}$ and 1 nmol mol$^{-1}$, respectively. According to the analyzer (Picarro G2401) specification sheet, the

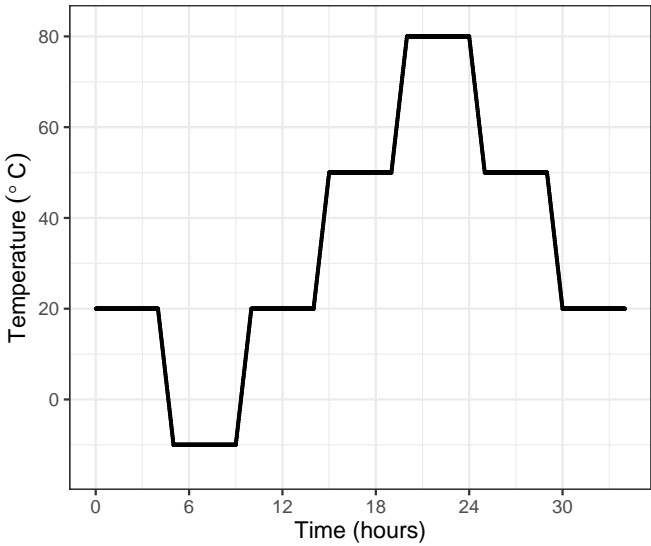

**Figure 3.** Temperature cycle set at the climate cabinet

5-minute, 1-$\sigma$ precision of the instrument is <1.5 nmol mol$^{-1}$ and <0.5 nmol mol$^{-1}$, whereas the 5-minute standard deviation of measured data corresponded to 5 nmol mol$^{-1}$ and 0.2 nmol mol$^{-1}$, for CO and CH$_4$, respectively. Therefore, we have concluded that no significant change was observed in the final amount fractions for any of the materials during the course of the pressure experiments for the species CO and CH$_4$. For H$_2$O, steel with mass flow controller and Dursan® loading showed
5 significantly higher maximal amount fractions, corresponding to about 3 times higher enhancements than other species.

In order to highlight the changes during the emptying of the measurement chamber, we show differences of the measured amount fractions from the initial amount fraction ($\Delta$CO$_2$ and $\Delta$H$_2$O) with respect to pressure for each material loading (Fig. 5). The first panel shows all materials together, whereas in the second and the third panels, individual runs are grouped together for each material loading. As indicated in Fig. 2, we made some changes to the measurement line in order to distinguish
10 whether various equipment upstream of the analyzer had an influence on the measurements. Therefore, for the steel loading, we show results of the pressure experiments with a mass flow controller, and a multiport valve. For CO$_2$, only Dursan® showed a significant difference as high as 1.85 $\pm$ 0.14 μmol mol$^{-1}$ in the final amount fractions. The enrichment in the CO$_2$ measurements started significantly earlier, and followed a distinctly different evolution compared to all other tested materials. We do link this enhancement to desorption from the surface of the material. Besides being resistant to adsorption of corrosive
15 and reactive media, the coating layer consists of amorphous silicon, oxygen and carbon (Silcotek Corporation). The desorption from the material to the gas mixture is most probably a combination of both desorption of newly adsorbed molecules after the filling, and desorption of already existing carbon in form of CO$_2$ on the coating. In order to distinguish this difference, fillings containing no CO and CO$_2$ such as synthetic air or N$_2$ would be worthwhile.

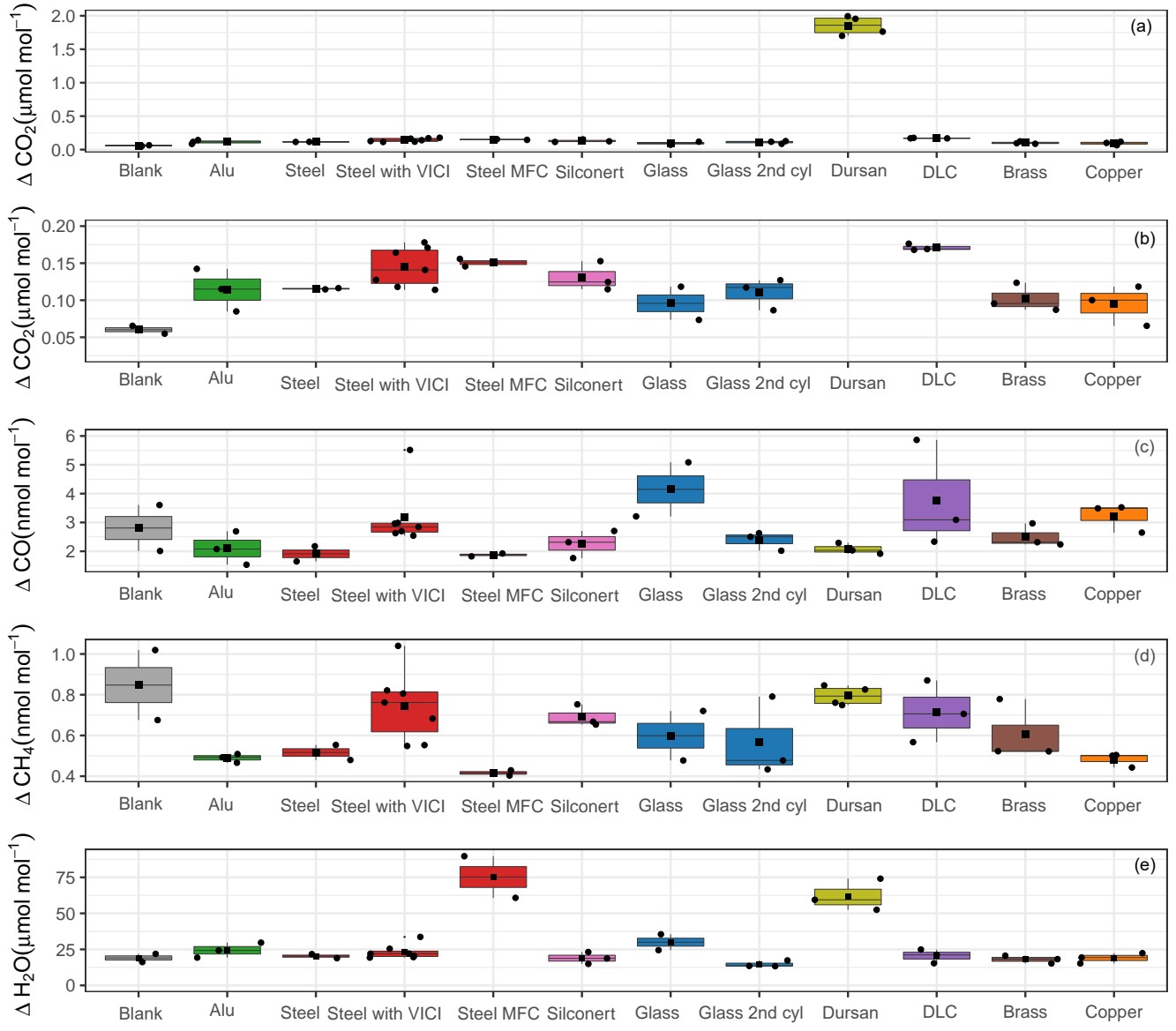

**Figure 4.** Box plots for all materials for the species **(a)** $CO_2$, **(b)** zoom-in for $CO_2$, **(c)** CO, **(d)** $CH_4$ and **(e)** $H_2O$. y-axes show the maximal amount fraction difference relative to the initial amount fraction. Horizontal lines in each box plot shows the median, whereas the square in the center of the box is the mean of the maximal amount fractions of the replicates.

For $CO_2$ measurements, the amount fraction differences for all materials except Dursan® were less than 0.17 µmol mol$^{-1}$, with slight differences among the various loadings. Of this difference, 0.05 µmol mol$^{-1}$ is related to the blank cylinder (background effect). The blank cylinder corresponded to the "14 bar after heating" case presented in Satar et al. (2019). More

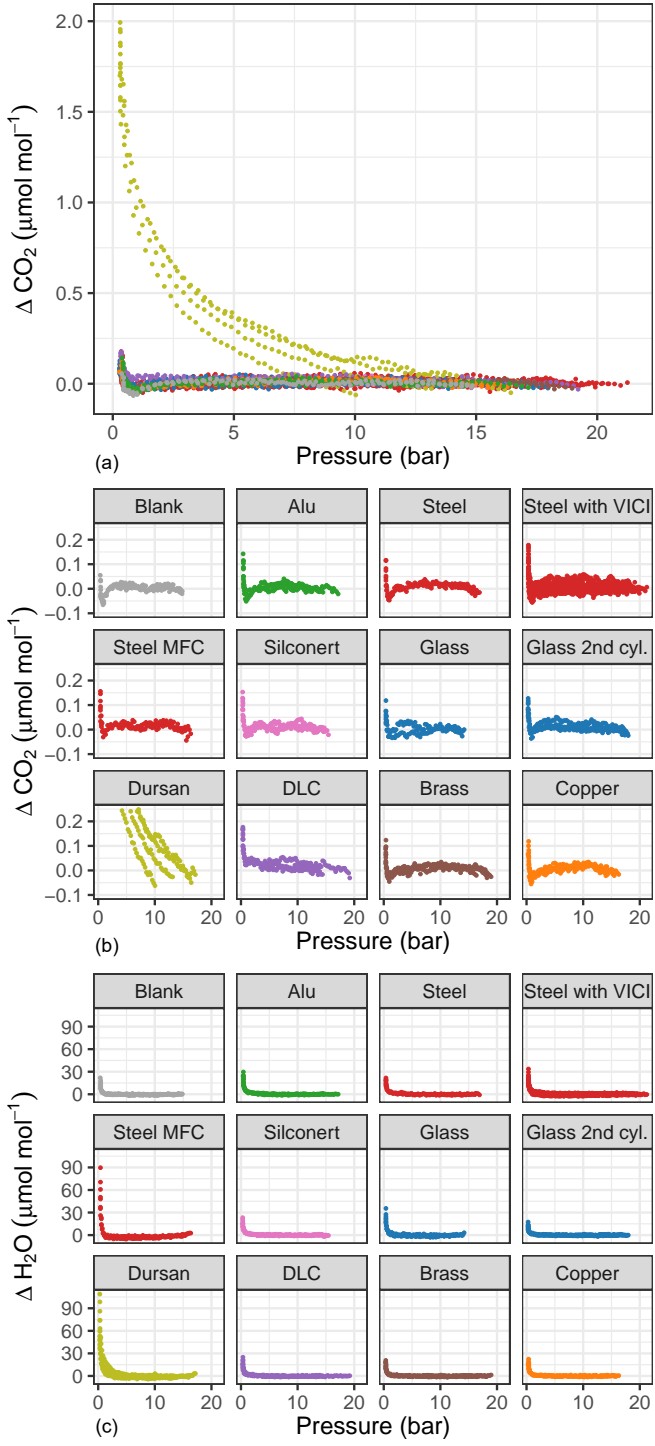

**Figure 5.** Amount fraction difference relative to the start of the experiment for **(a, b)** $CO_2$, and **(c)** $H_2O$ with respect to pressure for all tested materials. The first panel shows all materials together, whereas in the second and third panels, each material is plotted separately. Consistent color codes are used throughout the study.

information on the blank cylinder and its filling history is provided in the above-mentioned publication. It is also crucial to consider that during all material block experiments, glass pieces were also present in the small measurement chamber. When the material runs were compared to the experiments with glass, except the DLC loading, the remaining differences were in the order of 0.02 µmol mol$^{-1}$, which corresponded to the 5-minute standard deviation of the measured data. Moreover, during the evacuation of the measurement chamber with the DLC loading, a slightly increasing trend of –0.004 µmol mol$^{-1}$ bar$^{-1}$ was observed. For the steel loaded cylinder, the experiments where a multiport valve was upstream of the analyzer showed slightly more variation both for final amount fractions and during the pressure run. Whereas, the mass flow controller employed did not have a significant effect on the $CO_2$ measurements.

$H_2O$ measurements (Fig. 5) showed greater differences than $CO_2$ measurements, corresponding to 19.05 $\pm$ 2.84 µmol mol$^{-1}$ for the blank cylinder. Compared to other materials, Dursan® loading and the measurements with the mass flow controller showed significantly higher final amount fractions of 73.71 $\pm$ 12.55 µmol mol$^{-1}$, and 75.22 $\pm$ 14.45 µmol mol$^{-1}$, respectively. The difference observed for the mass flow controller was most probably related to a memory effect combined with teflon tape, since it was used for humid air prior to these measurements. Similar to the $CO_2$ response of Dursan® loading, the increase in $H_2O$ amount fraction is most probably a combination of both desorption of newly adsorbed molecules and, desorption from the coated layer. It is unlikely that the enrichment of $H_2O$ is related to the mother mixture since all other materials resulted in significantly lower amount fraction differences.

Since CO and $CH_4$ (Fig. 4c-d) showed no distinct differences in amount fractions with decreasing cylinder pressure, we include their analog plots in Fig. A1 for completeness. Under these "extreme" conditions of cylinder evacuation, the observed effects were minimal for most of the investigated materials. However, the Dursan® loading showing a difference revealed that the measurement chamber and the established procedure were successful to detect changes.

## 3.2 Temperature experiments

Based on the results of the pressure tests, the temperature experiments were conducted within a pressure range for which no pressure effect should occur, with the exception of Dursan®. In order to graphically distinguish the temperature effect on various materials, data was split into four different groups (Fig. 6): group 1 corresponded to materials showing the least response, group 2 were materials showing a significant temperature response, and group 3 and 4 corresponded to Dursan® and DLC separately, since they showed an order of magnitude higher differences for some of the measured species. Note that all x-axes in Fig. 6 correspond to a temperature cycle. Blank cylinder, glass, SilcoNert®2000 and brass loadings (Fig. 6a) showed the least response to temperature variations between –10 °C and 80 °C. For $CO_2$, the observed mean differences per material were less than 0.05 µmol mol$^{-1}$. However, this difference was as high as 11 nmol mol$^{-1}$ for CO. At 80 °C, there was a clear change in the amount fractions of CO, whereas this step change was not as clear in $CO_2$ measurements. The reason of this behavior is most probably related to the cylinder itself, which points to an irreversible chemical reaction, since the enhancement in amount fraction stayed prominent even when the cylinder was cooled back down to 20 °C. For $CH_4$, temperature variations resulted in non-significant amount fraction differences, and they stayed in a narrow range from –0.75 nmol mol$^{-1}$ to 0.5 nmol mol$^{-1}$. All material loadings showed an effect with respect to $H_2O$ measurements. This effect was reversible: lower $H_2O$ amount

fractions were observed at colder temperatures, and higher amount fractions at higher temperatures. This is an indication of adsorption/desorption, in which at colder temperatures desorption rate is lower, and the system equilibrates at lower amount fractions in the gas mixture. As the temperature increases, desorption rate increases, resulting temporally in higher amount fractions in water vapor. For the first group of materials, $H_2O$ measurements were within the range of –5 µmol mol$^{-1}$ to 5 µmol mol$^{-1}$. After cooling the cylinder to 20 °C, a slight change of 1 µmol mol$^{-1}$ compared to the initial amount fraction was observed.

Figure 6b shows steel, aluminum and copper. These loadings showed a more significant temperature response compared to group 1. Note that the dashed lines represent the ranges from the first group of materials. For $CO_2$, all materials showed a clear increase when the temperature was increased to 80 °C. This increase corresponded to $0.16 \pm 0.02$ µmol mol$^{-1}$, $0.10 \pm 0.01$ µmol mol$^{-1}$, and $0.05 \pm 0.02$ µmol mol$^{-1}$, for copper, steel, and aluminum respectively. After cooling back to 20 °C, the amount fraction increase in aluminum and steel dropped back to less than 0.07 µmol mol$^{-1}$, whereas for copper the difference persisted and was $0.13 \pm 0.02$ µmol mol$^{-1}$. For CO, the effects were even more significant. At 80 °C, $\Delta$CO for the copper loading increased to $29 \pm 1$ nmol mol$^{-1}$, and was followed by $16 \pm 4$ nmol mol$^{-1}$ and $14 \pm 2$ nmol mol$^{-1}$ for aluminum and steel respectively. Aluminum and steel loadings reached their maximum increase at 50 °C after cooling down from 80 °C, and stayed at that level even with further cooling. The increase in CO amount fraction for the copper loading continued and reached $43 \pm 1$ nmol mol$^{-1}$. This might be an an indication of an irreversible chemical reaction happening after the threshold of 80 °C which uses copper as a catalyst. This is further supported by the fact that the amount fraction enhancement after each temperature cycle was reproducible. Similarly to the first group, $CH_4$ measurements of the second group stayed in the narrow window of –0.75 nmol mol$^{-1}$ to 0.5 nmol mol$^{-1}$. For $H_2O$, group 2 materials showed a slightly greater effect than group 1 materials with a mean of $7.55 \pm 2.88$ µmol mol$^{-1}$ at 80 °C, and reached $11 \pm 1$ µmol mol$^{-1}$ for aluminum. After cooling to 20 °C, a difference over 2 µmol mol$^{-1}$ compared to the beginning was observed.

Figure 6c-d show Dursan® and DLC on separate panels for each of the measured species. For $CO_2$, Dursan® showed differences as high as $0.64 \pm 0.02$ µmol mol$^{-1}$. A fraction of this difference was related to the pressure decrease of the cylinder from 15 bar to 5 bar. However, during most of the temperature cycle including measurements at 80 °C, the pressure in the cylinder was over 9.5 bar, corresponding to a pressure effect of less than 0.25 µmol mol$^{-1}$. Even when this difference was considered, the temperature effect of this material was an order of magnitude greater than group 1 materials at 80 °C. Whereas at 20 °C at the end of the temparature cycle, the pressure contribution was as high as 0.5 µmol mol$^{-1}$. CO and $CH_4$ did not show any significant difference in their response compared to group 1 and group 2 materials. $H_2O$ measurements were higher than in the other groups, but reversible.

The DLC loading clearly showed different temperature response compared to all other materials, especially with regard to the variability of its replicates. $CO_2$ showed a mean difference of $0.15 \pm 0.04$ µmol mol$^{-1}$ compared to the beginning of the cycle. For CO and $CH_4$, the differences from the starting amount fractions were 10 and 20 times larger than the differences observed for other tested materials. At each temperature cycle, the measured CO difference at 80 °C decreased, for the first and last cycle, this difference corresponded to 87 nmol mol$^{-1}$ and 51 nmol mol$^{-1}$, respectively. This feature was observed in $CH_4$ and $H_2O$ measurements as well. During the first temperature cycle at 80 °C, $CH_4$ and $H_2O$ showed an increase of 18

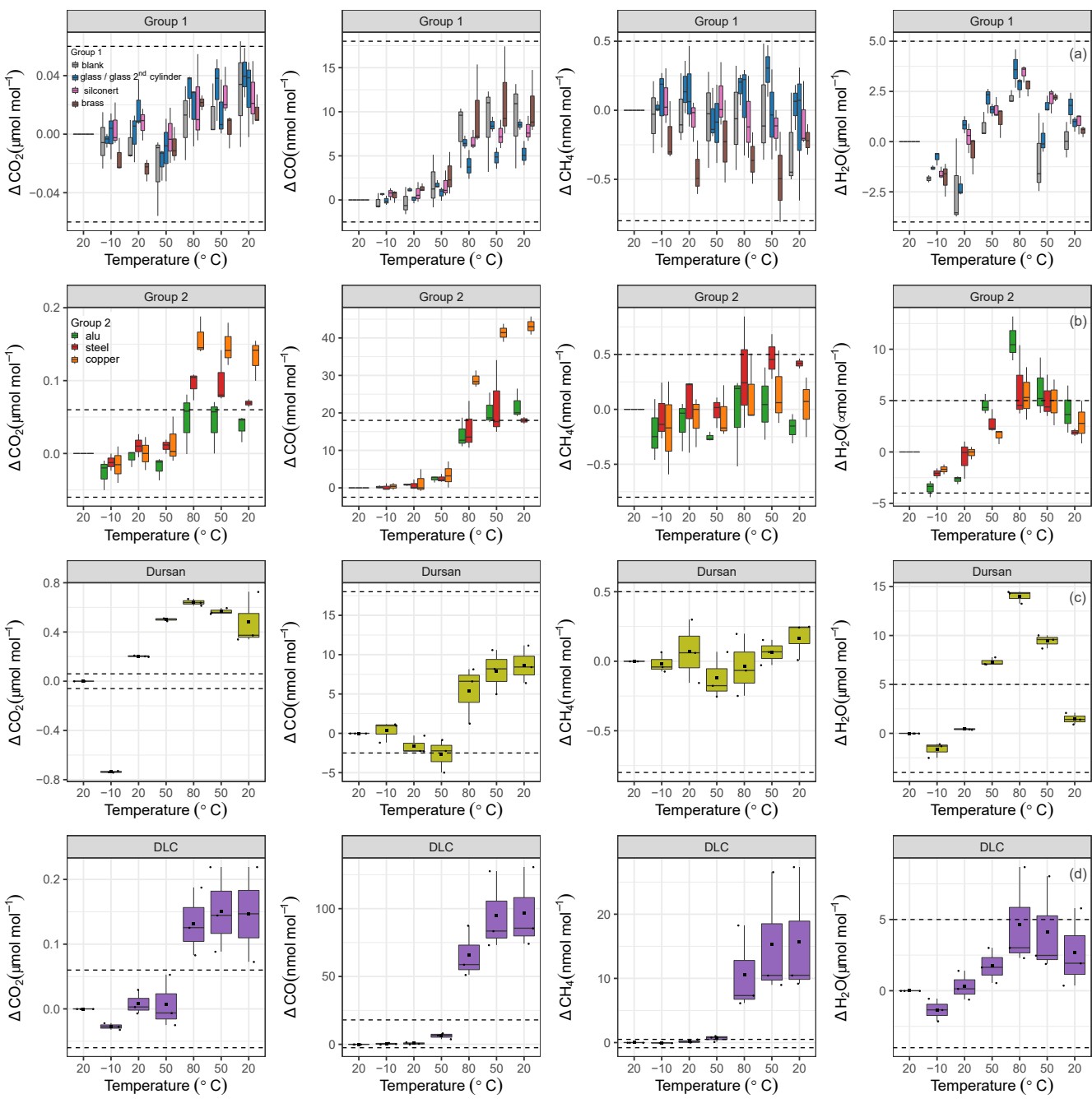

**Figure 6.** Temperature experiments grouped according to temperature response. **(a)** Group 1 materials are blank cylinder (gray), glass (blue), SilcoNert®2000-coated steel (pink), and brass (brown). **(b)** Group 2 materials are aluminum (green), stainless steel (red), and copper (orange). **(c)** Dursan® (light green) and **(d)** DLC (purple) coatings on stainless steel. Dashed lines indicate the same ranges for each species. The x-axes correspond to the temperature cycles (cf. Fig. 3), and the y-axes show the amount fraction differences relative to the first measurement block at 20 °C.

nmol mol$^{-1}$ and 8 µmol mol$^{-1}$, whereas during the third temperature cycle, CH$_4$ and H$_2$O showed an increase of 6 nmol mol$^{-1}$ and 3 µmol mol$^{-1}$, respectively. This behavior in DLC measurements showed that the underlying reason of the enhancement in the amount fractions were substances on the surface of the DLC coating, which by repeated heating were depleted. According to Grill (1999), thermal relaxation of the DLC film may occur at temperatures as low as 100 °C.

Moreover, it should be noted that during the set of the measurements presented in this study, the aluminum cylinder experienced the temperature cycle 30 times. This presumably resulted in a change of the background effect over the course of the presented analysis in the range of 0.04 µmol mol$^{-1}$. We measured the blank cylinder at the very beginning and at the end of the material set. The blank cylinder results shown in the Fig. 6a is a mean of the former and latter temperature experiments, therefore resulted in a higher variation. Due to the observed variation, we did not subtract the background to avoid disturbing other measurements. More detailed information on heating experiments and its consequences were presented in Satar et al. (2019).

## 4 Discussion

The presented setup enabled the investigation of surface effects under "extreme" conditions which favored adsorption/desorption. Compared to common usage in the atmospheric measurement and gas metrology communities, our study has differed in cylinder size, geometric surface to volume ratios, pressure and temperature ranges. Previous studies (Leuenberger et al., 2015; Brewer et al., 2018; Schibig et al., 2018) investigating surface effects in compressed gas cylinders have used (50 L, 10 L, or 29.5 L) cylinders. The geometric surface of the small (5 L) aluminum cylinder used in this study is 0.18 m$^2$, which results in a surface to volume ratio of 35.7 for the unloaded cylinder. Compared to 29.5 L Luxfer cylinders, the small cylinders are estimated to be more prone to adsorption by 40 %. Inserting material blocks into the aluminum cylinder further increased the surface area. Therefore, the setup allows to test materials under increased surface to volume ratios in which the surface effects should be stronger and dominant. However, despite our efforts of increasing the surface material effects were minor.

In addition to the properties of the materials, pressure and temperature play a role on surface effects. The following assumption lies behind the pressure experiments: if the material has adsorbed a significant amount of gas while filling the cylinder, this should be desorbed towards the end of the experiments controlled by desorption. The onset of the desorption for all tested materials except Dursan® and partly DLC was observed well-below atmospheric pressures.

Increasing temperature is expected to facilitate desorption by providing the required energy to desorb the gas molecules from the surface and mix into gas phase. On the contrary, cooling the cylinder and its content favor adsorption and it is expected that this results in a decrease in the measured amount fraction.

Testing pieces cut from the aluminum and steel cylinders commonly used in the community would be a valuable addition to enable direct comparison between the commonly used cylinder materials and the produced material blocks at low pressures and high temperatures. Moreover, in order to observe significant surface effects, materials of very high surface areas can be inserted into the measurement chamber. Some ideas would be using thin metal plates, metal spheres or metal pieces resulting from manufacturing processes (e.g. metal chips).

# 5 Conclusions

We have presented the pressure and temperature dependent response of the species $CO_2$, $CH_4$, CO and $H_2O$ with respect to glass, aluminum, copper, brass, steel, and three different commercially available coatings on stainless steel (Dursan®, SilcoNert®2000 and DLC). For the pressure response, under the circumstances of our experimental setup and procedure, within the pressure range varying from 15 bar to 200 mbar absolute, we were only able to detect changes for $CO_2$ in the loading with Dursan® coated stainless steel reaching 2 µmol mol$^{-1}$ enrichment in the amount fractions. All other materials showed effects less than 0.17 µmol mol$^{-1}$ for $CO_2$. Under exposure to temperatures from –10 °C to 80 °C, the response of glass, brass and SilcoNert®2000 coated steel were minimal, whereas DLC and Dursan® showed distinctly different temperature effects than all other tested materials. For most materials, including stainless steel, copper, aluminum, DLC and Dursan® a step change in the measured amount fractions were observed after 80 °C.

These experiments show that all coatings not necessarily enable more passive surfaces, and might result in enhancements when exposed to pressure and temperature changes. Materials currently used by the atmospheric measurement community for storing standard gases are well suited under 80 °C, which are typical utilisation temperatures. Moreover, the materials presented in this study are not only relevant for measurements of standard gases, but also of interest for other gas handling and measuring applications.

## Appendix A

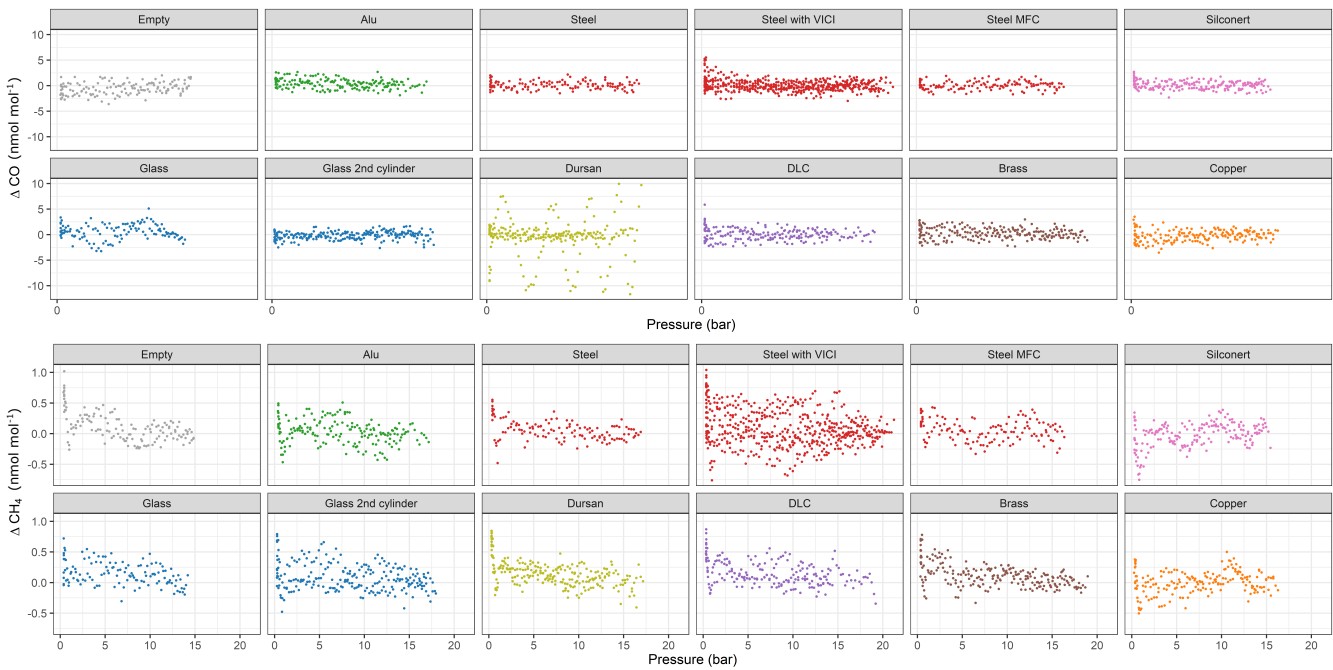

**Figure A1.** Amount fraction differences compared to the start of the experiment for CO, and CH$_4$ for all tested materials

*Acknowledgements.* This project is supported by a research contract (F-5232.30052) between the Swiss Federal Institute of Metrology (METAS) and the University of Bern as well as the SNF project "Klima- und Umweltphysik: Isotope im Erdklimasystem" (icoCEP)(SNF-200020_172550). The authors would like to thank to the Workshop of University of Bern for the production of the cylinders, and METAS Gas Analysis Laboratory and METAS workshop for their technical support during this work.

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
