# Peer review of "Towards an understanding of surface effects: Testing of various materials in a small volume measurement chamber and its relevance for atmospheric trace gas analysis"

_Atmospheric Measurement Techniques, 2019_

## Referee Comment (RC1) · Anonymous Referee #2 · 26 Aug 2019

**1   General comments**

This manuscript deals with adsorption/desorption of trace gases in air on various metal surfaces. While many existing studies have focused on real cases, testing types of cylinders in use in the atmospheric measurement community, this manuscript on the contrary describes experiments performed using specifically designed test cylinders, filled with an air mixture and various materials, to study potential adsorption phenomenon of gases ($CO_2$, $CO$, $CH_4$, water vapour) on the tested surfaces. The results

presented in this manuscript represent many hours of preparation and measurement, and are surely of value for the community measuring trace gases in the atmosphere and preparing reference gas mixture for this purpose. In particular, this study reports detectable and quite large effects for the coating Dursan for $CO_2$, which was unexpected, Dursan being advertised as a passivation treatment. Irreversible alteration of the amount fraction for most species and materials at temperatures equal or above 80°C are also reported. In many other cases, no clear adsorption/desorption effect can be seen, which is at the same time a bit disappointing for process analysis but also good news from the user's point of view. The manuscript is well organised and figures in particular have been prepared with great care and display the results very clearly. Some descriptions in the text may gain in clarity (suggestions hereafter under 'specific comments').

**2 Specific comments**

What is your method's limit of detection, i.e. the smallest adsorption/desorption effect that could be detected using the chosen measuring instrument? What does the thresholds of 0.2 $\mu$mol/mol you mention for $CO_2$ p. 9 l. 9, 6 nmol/mol for CO and 1 nmol/mol for $CH_4$ (p. 7 l. 15) represent? If these questions are answered in the companion paper, please cite it.

For the pressure tests in particular, very little adsorption/desorption effect is seen, making likely very hard to actually estimate a number of molecules adsorbed per unit of surface area and/or to compare with theoretical adsorption curves (even if, from the user's point of view, this is actually good news). This stated, it seems also clear that a new design allowing to cause larger adsorption effect would demand a substantial work and is beyond the scope of this manuscript. Still, how would you design a new test chamber / test material or how would you conceptually modify the present equip-

ment to provoke a larger effect that could then be better analysed? I would suggest to add a few lines discussing this in the discussion and/or conclusion.

Distinction container/content: I would suggest making clear reference to a gas mixture when writing of measuring, spiking or being adsorbed (e.g.: working gas, mother mixture), and to a gas container when writing of evacuating, cleaning, connecting, etc. (e.g.: working cylinder, mother cylinder). A few examples:

p. 3 l. 29: The fillings were done using compressed air from high pressure 50 l aluminium cylinders (LUX3586 and LUX 3575).

p. 3 l. 29-30: These two cylinders are called the mother cylinders and their air content the mother mixture from here on.

p.3 l. 31: In addition to the mother mixture, another mixture of comparable content and from a cylinder of comparable material and equipment to the mother cylinder was measured [...].

p. 3 l. 33: This mixture (from cylinder LUX3579) is refereed to as the working gas.

p. 4 l. 2-4: [...], the mother mixtures we spiked [...] using another compressed air mixture as carrier gas.

Please check that this distinction is clear through the manuscript.

p.5 l. 6: 'empty cylinder': it is stilled filled using the mother mixture so it is not empty strictly speaking. Maybe 'blank cylinder' (with the same meaning as 'blank measurement') would be more precise. Please modify through the manuscript (text, Tables, Figures).

p. 7 l. 12: 'end amount fraction': I would suggest replacing by 'final amount fraction'. Please check through the text.

**3 Technical corrections, phrasing**

Abstract: a direct mention of adsorption right at the beginning would be more clear. Suggestion: A critical issue [...] employed. Both measuring and preparing reference gas mixtures for trace gases are challenging due to e.g. adsorption/desorption of the substances of interest on surfaces; this is particularly critical at low amount fraction and/or for reactive gases. Therefore, to ensure [...]. This study focuses on testing potential adsorption/desorption effects for different materials [...].

Abstract l. 10: [...] to investigate the pressure dependency of adsorption up to 15 bar, and its temperature dependency [...].

p. 1 l. 18, suggestion: In order to achieve a high level of compatibility for data obtained at different sites and/or at different time, the World Meteorological Organisation [...].

p.1 l. 22: [...] but also by limiting any cause of molar fraction alteration.

p. 1 l. 22: maybe mention an order of magnitude for the lifetime of a standard cylinder?

p. 2 l. 8: larger volume

p. 2 l. 18: we aim at distinguishing these effects

p. 2 l. 28: on various surfaces.

p. 2 l. 30: According to the current literature,

p. 2 l. 34: the adsorption loss on the stainless steel surface

p. 3 l. 14: [...] we used the aluminium cylinder only.

p. 3 l. 17: [...] used in the atmospheric measurement community. This custom-made [...]

p. 5, legend of Fig. 2: related to the cleaning procedure

p. 7 l. 2: For the data analysis, for each temperature step the first 10 minutes of the measurements were not included in order to allow time for equilibration; the mean of the remaining 25 minutes was calculated.

p. 9 legend of Fig 5: whereas in the second and third panels

p. 9 l. 10 For example, [...] analyser showed [...] pressure run, whereas the mass flow [...].

p. 10 l. 6: Based on the results of the pressure tests, the temperature experiment were conducted within a pressure range for which no pressure effect should occur, [...].

p. 10, legend of Fig. 6: The x-axes correspond to the temperature cycles (cf. Fig. 3),

p. 10, legend of Fig. 6: does the y-axis show the amount fraction differences relative to the first measurement bloc done at 20°C? (There are three measurement blocs done at 20°C.)

p. 10 l. 7: In order to graphically distinguish [...]

p. 11 l. 9: remove 'Please' (check through the text).

Suggestion: displaying Fig. 6 and 7 on the same page.

---

## Referee Comment (RC2) · Anonymous Referee #3 · 3 Sep 2019

Atmos. Meas. Tech. Discuss.

**Review of the submitted article:**

**Towards an understanding of surface effects: Testing of various materials in a small volume measurement chamber and its relevance for atmospheric trace gas analysis**

Ece Satar, Peter Nyfeler, Céline Pascale, Bernhard Niederhauser, and Markus Leuenberger

General comments:

This paper describes a series of experiments aimed at comparing the adsorption of some atmospheric trace gases in various materials. Experiments were planned in a very structured way to allow meaningful observations. The study was part of a larger study on adsorptions, and it is clear that authors chose to limit this paper to one main variable: the surface material. A good number of different materials were chosen, and all of them appear to be of interest to the community. The paper is generally well written, well-structured, clear, and provides a number of details on the instruments and methods, with some further details missing.
However the discussion part of the paper is quite limited. The results need to be put in perspective with other published work, in particular on $CO_2$ with aluminium. It also misses explanation and assumptions on the phenomena at work. Previous work by Leuenberger included a complete model with an attempt to fit the results during similar experiment in large cylinders. This paper should at least summarise this effort and explain if such attempt was also made here, and why it does not appear.
Considering the type of comments provided below, I recommend a major revision before the paper can be published.

Specific comments by section:

Section 1. Introduction:
- Cylinders volume in this study compared to others: the introduction mentions this difference but does not state the potential impact on observations. In Schibig 2018 it is explained that cylinders smaller than 30 L should present larger effects, due to the surface to volume ratio. This should be observed and reflected through the introduction and the rest of the paper, in terms of the magnitude of observed effects compared to cylinders commonly used as standards.

Section 2.1:
- Small chambers in aluminium and steel cylinders were designed, but this study only reports observations with the aluminium cylinder. The rationale behind this choice should be added. Was it after the conclusions of the other paper?

- The analyser is mentioned line 27 without a description. Please add the model and the performances which are relevant to the study. In particular one needs to pay attention to the sensitivity for the compounds measured, to demonstrate that observations are meaningful (or not). The claimed repeatability of the instrument appears sometimes on the same order as the changes measured during the study.

- Compressed air used for the study: more details on the composition are clearly needed, at least nominal values provided by the company. The water content in particular is under question, as some of the observed differences are of the order of 70 μmol/mol. Does this mean the water amount fraction was even higher than this? This is important to clarify, considering that the work of Brewer et al. showed how water can be preferably adsorbed on surfaces, decreasing the adsorption of other compounds such as $CO_2$.

Section 2.2:

- Flow rate: previous studies of Schibig et al. and Brewer et al. mentioned an influence of the flow rate at which cylinders are being emptied. How was this taken into account? What was the flow rate during the measurements? Some consideration on this point should be provided.

- Pressure values during temperature studies: table 1 shows that the pressure could change when changing the temperature. Consider explaining the reason and potential impact on the results.

- It is explained that in this study, all reported values are in amount fraction difference. It can be assumed that this choice was made to plot all data together and be able to compare different observations. This might be a good reason, but absolute values should also be provided, at least once, to be able to compare the results in this study with others.

- Temperature cycle: please clarify that the container was refilled at the beginning of each new temperature step. This information could also be added on figure 3.

Section 3:

- It is said several times that changes observed with $CO_2$ are significant only for Dursan. However differences of the order of 0.15 µmol/mol were observed with other materials and this is comparable with the compatibility goal within GAWG. In other studies similar changes were not considered negligible. Some analysis in view of already published work should be added and made more consistent.

- The "empty" case needs further clarification. First on the term itself which is badly chosen as the container is always filled with gas. Second on the values compared to the other paper of the authors. They are apparently those of the case "aluminium, filled at 14 bar, after heating". This should be clarified and related to the choice of this material (best results?).

- The "steel" results can be confusing when compared to the other paper, where a difference of 0.5 µmol/mol was observed. The setup is of course not the same, but this would need some consideration and some assumptions to explain those discrepant results.

- Results on water: figure 4 shows up to 60 µmol/mol changes, which would mean quite large water content to start with. Was it the case? If not, where does the water come from?

Comments on figures:

Figure 5: consider splitting in different figures to allow a larger front. This is currently far too small.

Figures 6: the *x*-axes is very disturbing, even after the highlight in the text noting that it indicates the temperature cycle, which is why the scale is not linear. There is some logic in this choice, but it discards the possibility to clearly see the temperature effect. Consider plotting with a linear temperature scale using a color code or different shapes to show the time. Another option is to use time as x-axes and superpose the temperature cycle.

---

## Referee Comment (RC3) · Anonymous Referee #1 · 10 Sep 2019

This paper described the testing of various materials in an aluminum chamber designed such that various materials can be inserted in the chamber and tested for adsorption effects. This paper is a follow-on from a previous paper that described the testing chamber and analytical challenges (10.5194/amt-2019-197). The paper is well-written and contains informative figures. For many of the materials tested, desorption of $CO_2$, $CH_4$, and CO were minimal, which is good news for the measurement community. However, it is not exactly clear how the results of these experiments relate to atmospheric trace gas analysis due to significant differences between this work and

how compressed gas standards are used in practice.

General comments:

You mention that you did not subtract the background, or blank (empty) run from the experiments. In figure 5, it would seem that the empty run shows much the same signals as the materials tested, which I suppose is why you state that only Duran showed significant difference. I think this needs further explanation related to the other Satar et al 2019 paper (10.5194/amt-2019-197). On page 12, you mention that you measured the "blank" or background at the beginning at at the end of the experiments, and the the "blank" shown in figure 6 is the mean. How for this impact the conclusions? Do you know if the background changed smoothly over time, or abruptly as a results of adding Dursan or DLC? Further, I think your can better explain why you chose the maximum deviation (page 7, line 9), given that making measurements at sub-ambient pressure introduces complications, as described in the other Satar et al 2019 paper.

How do these results compare with others studies (Leuenberger et al 2015; Schibig et al 2018) that suggest that a Langmuir isotherm can be used to model the physical adsorption? Your tests seem to show a much steeper increase at the lowest pressures. Is the abrupt increase at the lowest pressure related to desorption, or is it complicated by analysis at low pressure? I realize the analysis is discussed in the the first Satar et al 2019 paper, but I think some important aspects need to be repeated here.

Since it not common to use a cylinder of gas down to less than 10% of the fill pressure, what would these results say about that practice?

I am also concerned about how to interpret the results with respect to materials used by the scientific community. The inside of an aluminum cylinder, for example, does not typically look like the outside. Presumably the process of manufacturing a cylinder (using a ram rod) alters the surface roughness of the inside, and may deposit trace elements on the internal surface. While you tested the same aluminum alloy used for cylinders, the tested materials might not be representative of actual cylinders. Have

you tested pieces of material cut from aluminum or steel cylinders? Maybe it doesn't matter since the results don't show significant desorption, but this should at least be discussed.

Specific Comments

P1, L5: suggest: "For this study we used small-volume chambers designed to be used for adsorption studies".

P3, L30: suggest: "A mother cylinder was . . ."

Figure 4: Minor point: I'm not sure of the significance of the box plots based on only 3 data points. I guess you are assuming normal distributions and assigning quartiles based on the standard deviation.

---

## Author Comment (AC1) · 21 Oct 2019

**Reply to the review of Anonymous Referee #2**

The authors would like to thank Anonymous Referee for the valuable comments. In the following, referee's comments are given in bold and author's responses in plain text. Suggested new text is quoted in italics together with page and line numbers.

**1 General comments**

**This manuscript deals with adsorption/desorption of trace gases in air on various metal surfaces. While many existing studies have focused on real cases, testing types of cylinders in use in the atmospheric measurement community, this manuscript on the contrary describes experiments performed using specifically designed test cylinders, filled with an air mixture and various materials, to study potential adsorption phenomenon of gases ($CO_2$, CO, $CH_4$, water vapour) on the tested surfaces. The results presented in this manuscript represent many hours of preparation and measurement, and are surely of value for the community measuring trace gases in the atmosphere and preparing reference gas mixture for this purpose. In particular, this study reports detectable and quite large effects for the coating Dursan for $CO_2$, which was unexpected, Dursan being advertised as a passivation treatment. Irreversible alteration of the amount fraction for most species and materials at temperatures equal or above 80 °C are also reported. In many other cases, no clear adsorption/desorption effect can be seen, which is at the same time a bit disappointing for process analysis but also good news from the user's point of view. The manuscript is well organised and figures in particular have been prepared with great care and display the results very clearly. Some descriptions in the text may gain in clarity (suggestions hereafter under 'specific comments').**

**2 Specific comments**

**What is your method's limit of detection, i.e. the smallest adsorption/desorption effect that could be detected using the chosen measuring instrument? What does the thresholds of 0.2 µmol/mol you mention for $CO_2$ p. 9 l. 9, 6 nmol/mol for CO and 1 nmol/mol for $CH_4$ ( p. 7 l. 15) represent? If these questions are answered in the companion paper, please cite it.**

We agree with our reviewer that this point needs further clarification. The word "threshold" is not correctly used in this context, therefore we replace it with the following phrasing on page 9 line 10:

*"For $CO_2$ measurements, the amount fraction differences for all materials except Dursan were less than 0.17 µmol mol$^{-1}$, with slight differences among the various loadings."*

For the presented experiments, we have used a Picarro G2401 CRDS analyzer. According to the specification sheet, the 5-minute, 1-σ precision of the instrument is <0.02 µmol mol$^{-1}$, <1.5 nmol mol$^{-1}$ and <0.5 nmol mol$^{-1}$, for $CO_2$, CO and $CH_4$, respectively. The numbers presented in our study (0.17 µmol mol$^{-1}$, 6 nmol mol$^{-1}$ and 1 nmol mol$^{-1}$, for $CO_2$, CO and $CH_4$, respectively) were reported to highlight that the observed changes have not exceeded these values but are higher than instrument precision stated in the specification sheet. These values can further be compared to the 5-minute standard deviation of measured data for $CO_2$, CO and $CH_4$ which in our case corresponded to 0.02 µmol mol$^{-1}$, 5 nmol mol$^{-1}$, and 0.2 nmol mol$^{-1}$, respectively. For $CO_2$ measurements, an explanation is already provided to clarify the significance of the observed changes within the reply to the anonymous referee#1. For CO and $CH_4$ considering the standard deviation of the measurements and the specifications of the analyzer, the observed maximal amount fractions are not significant and do not show material dependency.

We will add and rearrange the paragraph on page 7 line 13:

*"For CO and CH$_4$, the maximum difference in the amount fractions did not exceed 6 nmol mol and 1 nmol mol$^{-1}$, respectively. According to the analyzer (Picarro G2401) specification sheet, the 5-minute, 1-σ precision of the instrument is <1.5 nmol mol$^{-1}$ and <0.5 nmol mol$^{-1}$, whereas the 5-minute standard deviation of measured data corresponded to 5 nmol mol$^{-1}$ and 0.2 nmol mol$^{-1}$, for CO and CH$_4$, respectively. Therefore, we have concluded that no significant change was observed in the final amount fractions for any of the materials during the course of the pressure experiments for the species CO and CH$_4$."*

**For the pressure tests in particular, very little adsorption/desorption effect is seen, making likely very hard to actually estimate a number of molecules adsorbed per unit of surface area and/or to compare with theoretical adsorption curves (even if, from the user's point of view, this is actually good news). This stated, it seems also clear that a new design allowing to cause larger adsorption effect would demand a substantial work and is beyond the scope of this manuscript. Still, how would you design a new test chamber / test material or how would you conceptually modify the present equipment to provoke a larger effect that could then be better analysed? I would suggest to add a few lines discussing this in the discussion and/or conclusion.**

We thank our reviewer for appreciating the difficulty of estimating the number of adsorbed molecules. In order to increase adsorption, we would recommend inserting materials of very high surface areas into the measurement chambers. Some ideas would be using thin metal plates, metal spheres or metal pieces resulting from manufacturing processes (e.g. metal chips).

We will add the following statement at the end of the discussion:

*"Moreover, in order to observe significant surface effects, materials of very high surface areas can be inserted into the measurement chamber. Some ideas would be using thin metal plates, metal spheres or metal pieces resulting from manufacturing processes (e.g. metal chips)."*

**Distinction container/content: I would suggest making clear reference to a gas mixture when writing of measuring, spiking or being adsorbed (e.g.: working gas, mother mixture), and to a gas container when writing of evacuating, cleaning, connecting, etc. (e.g.: working cylinder, mother cylinder). A few examples:**

**p. 3 l. 29: The fillings were done using compressed air from high pressure 50 l aluminum cylinders (LUX3586 and LUX 3575).** – text modified accordingly

**p. 3 l. 29-30: These two cylinders are called the mother cylinders and their air content the mother mixture from here on.** – text modified accordingly

**p.3 l. 31: In addition to the mother mixture, another mixture of comparable content and from a cylinder of comparable material and equipment to the mother cylinder was measured [...].** – text modified accordingly

**p. 3 l. 33: This mixture (from cylinder LUX3579) is refereed to as the working gas.** – text modified accordingly

**p. 4 l. 2-4: [...], the mother mixtures we spiked [...] using another compressed air mixture as carrier gas.** – text modified accordingly

**Please check that this distinction is clear through the manuscript.**

**p.5 l. 6: 'empty cylinder': it is stilled filled using the mother mixture so it is not empty strictly speaking. Maybe 'blank cylinder' (with the same meaning as 'blank measurement') would be more precise. Please modify through the manuscript (text, Tables, Figures).**

We understand the concern of our reviewer, we have changed *"empty cylinder"* to *"blank cylinder"* throughout the manuscript.

**p. 7 l. 12: 'end amount fraction': I would suggest replacing by 'final amount fraction'. Please check through the text.** – "end amount fraction" was changed to "final amount fraction" in the manuscript.

**3 Technical corrections, phrasing**

**Abstract: a direct mention of adsorption right at the beginning would be more clear. Suggestion: A critical issue [...] employed. Both measuring and preparing reference gas mixtures for trace gases are challenging due to e.g. adsorption/desorption of the substances of interest on surfaces; this is particularly critical at low amount fraction and/or for reactive gases. Therefore, to ensure [...]. This study focuses on testing potential adsorption/desorption effects for different materials [...].**

We agree with our reviewer that an earlier mentioning of adsorption is essential, we have changed the text accordingly.

**Abstract l. 10: [...] to investigate the pressure dependency of adsorption up to 15 bar, and its temperature dependency [...].** – text modified accordingly

**p. 1 l. 18, suggestion: In order to achieve a high level of compatibility for data obtained at different sites and/or at different time, the World Meteorological Organisation [...].** – text modified accordingly

**p.1 l. 22: [...] but also by limiting any cause of molar fraction alteration. –** text modified to *"but also by limiting any cause of amount fraction alteration."*

**p. 1 l. 22: maybe mention an order of magnitude for the lifetime of a standard cylinder?**

On page 1 at line 22, the sentence is modified accordingly:

*"During their relatively long lifetime in the order of decades, standard gas cylinders […]"*

**p. 2 l. 8: larger volume** – text modified accordingly

**p. 2 l. 18: we aim at distinguishing these effects** – text modified accordingly

**p. 2 l. 28: on various surfaces.** – text modified accordingly

**p. 2 l. 30: According to the current literature,** – text modified accordingly

**p. 2 l. 34: the adsorption loss on the stainless steel surface** – text modified accordingly

**p. 3 l. 14: [...] we used the aluminium cylinder only.** – text modified accordingly

**p. 3 l. 17: [...] used in the atmospheric measurement community. This custom-made[...]** – text modified accordingly

**p. 5, legend of Fig. 2: related to the cleaning procedure** – text modified accordingly

**p. 7 l. 2: For the data analysis, for each temperature step the first 10 minutes of the measurements were not included in order to allow time for equilibration; the mean of the remaining 25 minutes was calculated.** – text modified accordingly

**p. 9 legend of Fig 5: whereas in the second and third panels** – text modified accordingly

**p. 9 l. 10 For example, [...] analyser showed [...] pressure run, whereas the mass flow[...].** – text modified accordingly

**p. 10 l. 6: Based on the results of the pressure tests, the temperature experiments were conducted within a pressure range for which no pressure effect should occur, [...].** – text modified accordingly

**p. 10, legend of Fig. 6: The x-axes correspond to the temperature cycles (cf. Fig. 3),** – text modified accordingly

**p. 10, legend of Fig. 6: does the y-axis show the amount fraction differences relative to the first measurement bloc done at 20∘C? (There are three measurement blocs done at 20∘C.)**

We thank our reviewer for pointing this out. We changed the legend accordingly:

*"the y-axes show the amount fraction differences relative to the first measurement block at 20 °C."*

**p. 10 l. 7: In order to graphically distinguish [...]** – text modified accordingly

**p. 11 l. 9: remove 'Please' (check through the text).** – text removed accordingly

**Suggestion: displaying Fig. 6 and 7 on the same page.**

We agree with our reviewer and combined the two figures into one.

[Figure]

**Figure 6.** Temperature experiments grouped according to temperature response. (a) Group 1 materials are blank cylinder (gray), glass (blue), SilcoNert®2000-coated steel (pink), and brass (brown). (b) Group 2 materials are aluminum (green), stainless steel (red), and copper (orange). (c) Dursan® (light green) and (d) DLC (purple) coatings on stainless steel. Dashed lines indicate the same ranges for each species. The x-axes correspond to the temperature cycles (cf. Fig.3), and the y-axes show the amount fraction differences relative to the first measurement block at 20 °C.

---

## Author Comment (AC2) · 21 Oct 2019

**Reply to the review of Anonymous Referee #3**

The authors would like to thank Anonymous Referee for the valuable comments. In the following, referee's comments are given in bold and author's responses in plain text. Suggested new text is quoted in italics together with page and line numbers.

**General comments:**

**This paper describes a series of experiments aimed at comparing the adsorption of some atmospheric trace gases in various materials. Experiments were planned in a very structured way to allow meaningful observations. The study was part of a larger study on adsorptions, and it is clear that authors chose to limit this paper to one main variable: the surface material. A good number of different materials were chosen, and all of them appear to be of interest to the community. The paper is generally well written, well-structured, clear, and provides a number of details on the instruments and methods, with some further details missing. However, the discussion part of the paper is quite limited. The results need to be put in perspective with other published work, in particular on $CO_2$ with aluminium. It also misses explanation and assumptions on the phenomena at work. Previous work by Leuenberger included a complete model with an attempt to fit the results during similar experiment in large cylinders. This paper should at least summarise this effort and explain if such attempt was also made here, and why it does not appear. Considering the type of comments provided below, I recommend a major revision before the paper can be published.**

We would like to thank our reviewer for the insights and opinions. As already stated by the reviewer, this study focuses on surface material. The previous study on these small cylinders (Satar et al. 2019, 10.5194/amt-2019-197) have already concentrated on comparing the newly built small cylinders with the existing literature. In the presented work, we aimed at going a step further and have used the aluminum cylinder as the measurement chamber. We'll include a separate discussion section (presented as well in the replies to anonymous referee #1). Regarding a model of fit for the results presented in this study, the observed maximal deviation from the initial amount fraction for the blank cylinder was as low as 0.05 µmol mol$^{-1}$ making modelling of this increase extremely difficult. For the material loadings, adsorbed amounts should be distributed between the blank cylinders and the material blocks. However, subtracting the maximal amount fraction difference of the glass loaded cylinder from the material loaded cylinder resulted in amount fractions in the order of the standard deviation of the measured data. Therefore, for the majority of the materials fitting the Langmuir isotherm is not reasonable (presented as well in the replies to anonymous referee #1).

Please see point by point comments for the comparison between this work and the previous studies.

**Specific comments by section:**

**Section 1. Introduction:**

**-Cylinders volume in this study compared to others: the introduction mentions this difference but does not state the potential impact on observations. In Schibig 2018 it is explained that cylinders smaller than 30 L should present larger effects, due to the surface to volume ratio. This should be observed and reflected through the introduction and the rest of the paper, in terms of the magnitude of observed effects compared to cylinders commonly used as standards.**

Despite our efforts of increasing the surface area the material effects were minor, the geometric surface to volume ratio in the material experiments were 71.4, whereas this ratio is only 25.4 in Schibig et al. (2018). Our study with various different materials have revealed that even by increasing the

surface areas the desorbed amount at the end of the experiment until sub-atmospheric pressures is not significant for the materials except Dursan and DLC.

In the discussion part the following paragraph will be added (already presented in the replies to anonymous referee#1):

*"The presented setup enabled the investigation of surface effects under "extreme" conditions which favored adsorption/desorption. Compared to common usage in the atmospheric measurement and gas metrology communities, our study has differed in cylinder size, geometric surface to volume ratios, pressure and temperature ranges. Previous studies (Leuenberger et al. 2015, Brewer et al. 2018, Schibig et al. 2018) investigating surface effects in compressed gas cylinders have used (50 L, 10 L, or 29.5 L) cylinders. The geometric surface of the small (5 L) aluminum cylinder used in this study is 0.18 $m^2$, which results in a surface to volume ratio of 35.7 for the unloaded cylinder. Compared to 29.5 L Luxfer cylinders, the small cylinders are estimated to be more prone to adsorption by 40 %. Inserting material blocks into the aluminum cylinder further increased the surface area. Therefore, the setup allows to test materials under increased surface to volume ratios in which the surface effects should be stronger and dominant. However, despite our efforts of increasing the surface material effects were minor."*

**Section 2.1:**

**-Small chambers in aluminium and steel cylinders were designed, but this study only reports observations with the aluminium cylinder. The rationale behind this choice should be added. Was it after the conclusions of the other paper?**

For the material experiments we have chosen to use the cylinder with the smallest background effect. We will add the following on page 3 line 14:

*"Since the aluminum cylinder showed smaller effects with respect to surface effects in the previous study (Satar et al, 2019), we have chosen to use the aluminum cylinder only for the material experiments in order to minimize the background effect related to the measurement chamber."*

**-The analyser is mentioned line 27 without a description. Please add the model and the performances which are relevant to the study. In particular one needs to pay attention to the sensitivity for the compounds measured, to demonstrate that observations are meaningful (or not). The claimed repeatability of the instrument appears sometimes on the same order as the changes measured during the study.**

We will mention the name of the analyzer earlier on page 3 line 27:

*"On the measurement line between the pressure regulator and the Picarro Cavity-Ring Down Spectroscopy analyzer (CRDS) G2401 either an electropolished stainless steel 1/4'' tubing ,…"*

On page 5 at line 3, we will add the following:

*"The experiments were conducted using a Picarro G2401 CRDS analyzer enabling measurements of $CO_2$, CO, $CH_4$ and $H_2O$. According to the specification sheet of the analyzer, 5 minute, 1-σ standard deviation is <0.02 μmol $mol^{-1}$, <1.5 nmol $mol^{-1}$, <0.5 nmol $mol^{-1}$ and <50 μmol $mol^{-1}$ for the species $CO_2$, CO, $CH_4$ and $H_2O$, respectively. In order to investigate the material's pressure dependency, the cylinder was filled through expansion from the mother cylinder to around 15 bar, and was evacuated through the Picarro analyzer."*

**-Compressed air used for the study: more details on the composition are clearly needed, at least nominal values provided by the company. The water content in particular is under question, as some**

**of the observed differences are of the order of 70 μmol/mol. Does this mean the water amount fraction was even higher than this? This is important to clarify, considering that the work of Brewer et al. showed how water can be preferably adsorbed on surfaces, decreasing the adsorption of other compounds such as $CO_2$.**

The observed differences of the 70 $\mu$mol mol$^{-1}$ were not related to the water content of the mother mixtures, but were related to the equipment or material involved in the experiments. This is already explained on page 9 at lines 14-18 for the runs with mass flow controller.

On page 4 at line 4 the following will be added:

*"After spiking the mother mixture, the composition of LUX3575 was 428.59 $\mu$mol mol$^{-1}$, 1083.73 nmol mol$^{-1}$, 2132.93 nmol mol$^{-1}$ and <15 $\mu$mol mol$^{-1}$ for $CO_2$, $CO$, $CH_4$ and $H_2O$."*

On page 9 at line 18, we will add the following:

*"Similar to the $CO_2$ response of Dursan loading, the increase in $H_2O$ amount fraction is most probably a combination of both desorption of newly adsorbed molecules and, desorption from the coated layer. It is unlikely that the enrichment of $H_2O$ is related to the mother mixture since all other materials resulted in significantly lower amount fraction differences."*

**Section 2.2:**

**-Flow rate: previous studies of Schibig et al. and Brewer et al. mentioned an influence of the flow rate at which cylinders are being emptied. How was this taken into account? What was the flow rate during the measurements? Some consideration on this point should be provided.**

Schibig et al. (2018) and Brewer et al. (2018) have conducted their measurements at high and low flow rates. In Schibig et al. (2018), low and high flow conditions were 0.3 L min$^{-1}$ and 5.0 L min$^{-1}$, whereas in Brewer et al. (2018) the low and high flow rates were 0.7 L min$^{-1}$ and 5.5 L min$^{-1}$, respectively. The flow rate in the presented experiments in this study as well as Satar et al. (2019) are comparable to the low flow conditions. In contrast to the above-mentioned previous studies, there was no excess flow prior to the analyzer. At the beginning of the experiment, the flow rate was 220 mL min$^{-1}$ (STP) and towards the end of the experiment it was 15 mL min$^{-1}$ (STP). More information on flow rate is included in Sect. 3.1.1 of Satar et al. (2019). Since we have conducted the measurements at low flow conditions, other fractionation effects due to a temperature gradient in the cylinder are not expected.

We will add the following flowrate information on page 6 line 1:

*"There was no flow regulation after the pressure regulator prior to the analyzer inlet. At the beginning of the experiment the flow rate was 220 mL min$^{-1}$ (STP) and towards the end of the experiment it decreased to 15 mL min$^{-1}$."*

**-Pressure values during temperature studies: table 1 shows that the pressure could change when changing the temperature. Consider explaining the reason and potential impact on the results.**

We think there was a misunderstanding in the interpretation of the values presented in Table 1. The three pressure values shown in the table only show the starting pressures of each experiment, and does not give information on pressure change related to temperature change. Regarding pressure changes during the temperature experiments, these changes can be estimated using the ideal gas equation. For example, for a filling of 15 bar pressure and 20 °C temperature a pressure of 18.1 bar at 80 °C and 13.5 bar at -10 °C is expected. At these ranges, no pressure effect is expected. This point is already taken into account and discussed for the temperature experiments (on page 10 line 6).

For clarification, we will add the following in the caption of Table 1:

*"The pressure values indicate the pressure in the small cylinder at the beginning of each replicate run."*

**-It is explained that in this study, all reported values are in amount fraction difference. It can be assumed that this choice was made to plot all data together and be able to compare different observations. This might be a good reason, but absolute values should also be provided, at least once, to be able to compare the results in this study with others.**

Indeed, similar to other studies we have preferred to plot our results in amount fraction differences. This approach enables to compare different observations and also highlights the measured differences. The mother cylinder content is compressed natural air. We have added the composition of the cylinder to section 2.1.

**-Temperature cycle: please clarify that the container was refilled at the beginning of each new temperature step. This information could also be added on figure 3.**

We think that there is a misunderstanding at this point. The cylinder was not filled at the beginning of each new temperature step. The cylinder was filled to about 15 bar at the beginning of the temperature cycle (Fig. 3) and refilled after a full temperature cycle.

**Section 3:**

**-It is said several times that changes observed with $CO_2$ are significant only for Dursan. However, differences of the order of 0.15 µmol/mol were observed with other materials and this is comparable with the compatibility goal within GAWG. In other studies, similar changes were not considered negligible. Some analysis in view of already published work should be added and made more consistent.**

We thank our reviewer for pointing this out. It should be noted that the differences observed in this study were observed at sub-atmospheric pressures, other studies including Leuenberger et al. (2015), Schibig et al. (2018) and Brewer et al. (2018) observed these differences at an earlier onset at higher pressures. In our opinion, this study should be seen independently from the existing literature due to the following reasons: (i) the experimental setup used in this study is not comparable to previously published work in terms of inserting different materials in a measurement chamber, (ii) an introduction into the blank cylinders and their comparison to existing literature is already presented in detail within the scope of Satar et al. (2019), (iii) the surface to volume ratios in the current study is increased on purpose to increase adsorption/desorption effects.

In the previous study in which the cylinders were introduced (Satar et al., 2019), a discussion on how the small cylinders behave in comparison to other studies has already been included for the blank cylinders. In our opinion the focus of the presented work is to understand the effects of different materials. Nevertheless, we suggest to add the following the paragraph to discussion (presented as well in the replies to anonymous referee #1):

*"The presented setup enabled the investigation of surface effects under "extreme" conditions which favored adsorption/desorption. Compared to common usage in the atmospheric measurement and gas metrology communities, our study has differed in cylinder size, geometric surface to volume ratios, pressure and temperature ranges. Previous studies (Leuenberger et al. 2015, Brewer et al. 2018, Schibig et al. 2018) investigating surface effects in compressed gas cylinders have used (50 L, 10 L, or 29.5 L) cylinders. The geometric surface of the small (5 L) aluminum cylinder used in this study is 0.18 m², which results in a surface to volume ratio of 35.7 for the unloaded cylinder. Compared to 29.5 L Luxfer cylinders, the small cylinders are estimated to be more prone to adsorption by 40 %. Inserting material*

*blocks into the aluminum cylinder further increased the surface area. Therefore, the setup allows to test materials under increased surface to volume ratios in which the surface effects should be stronger and dominant."*

Regarding the significance of 0.15 µmol mol$^{-1}$, we clarify as follows on page 9, at line 10 (presented as well in the replies to anonymous referee #1):

*"For $CO_2$ measurements, the amount fraction differences for all materials except Dursan were less than 0.17 µmol mol$^{-1}$, with slight differences among the various loadings. Of this difference, 0.05 µmol mol$^{-1}$ is related to the blank cylinder (background effect). The blank cylinder corresponded to the "14 bar after heating" case presented in Satar et al. (2019). More information on the blank cylinder and its filling history is provided in the above-mentioned publication. It is also crucial to consider that during all material block experiments, glass pieces were also present in the small measurement chamber. When the material runs were compared to the experiments with glass, except the DLC loading, the remaining differences were in the order of 0.02 µmol mol$^{-1}$, which corresponded to the 5-minute standard deviation of the measured data."*

**-The "empty" case needs further clarification. First on the term itself which is badly chosen as the container is always filled with gas. Second on the values compared to the other paper of the authors. They are apparently those of the case "aluminium, filled at 14 bar, after heating". This should be clarified and related to the choice of this material (best results?).**

In order to avoid this misunderstanding, we have changed *"empty"* to *"blank"* as suggested by the anonymous reviewer 2.

Aluminum cylinder was chosen as a measurement chamber for the presented study, since aluminum is the commonly used material in the atmospheric measurement community. Since all material experiments were conducted after the temperature experiments presented in Satar et al. (2019), we have naturally used the "aluminum cylinder after heating". The choice of the cylinder is already clarified above (Section 2.1), and the blank results are linked to the 14 bar after heating case in the discussion section (please see the suggestions above).

**-The "steel" results can be confusing when compared to the other paper, where a difference of 0.5 µmol/mol was observed. The setup is of course not the same, but this would need some consideration and some assumptions to explain those discrepant results.**

We thank our reviewer for his/her attention. The discrepancy between the two steel result is most likely related to the different composition of steel used in these two studies. In the presented study stainless steel blocks of (316L) are used, whereas the previous study uses a steel cylinder of hardened and tempered steel (1.7218 / 25CrMo4 EN AW-6061) This information is already presented at the respective papers.

**-Results on water: figure 4 shows up to 60 µmol/mol changes, which would mean quite large water content to start with. Was it the case? If not, where does the water come from?**

The two cases with high water changes were already explained in the manuscript on page 9 at line 14 and with the suggestions above. It is important to note that the big differences in the amount fractions are observed towards the end of the experiments where desorption is expected to be at play. We relate the high water content with the mass flow controller and the Dursan blocks. For the remaining runs, considering the low flow rate and the duration of the high water vapor content episodes, the

integrated amount of water vapor is reasonable since there is a trace amount of water available in the mother cylinder which will be adsorbed at high pressures.

**Comments on figures:**

**Figure 5: consider splitting in different figures to allow a larger front. This is currently far too small.**

We understand the reviewer's concern. We will increase the font size, but will keep all subfigures.

[Figure]

**Figure 5.** Amount fraction difference relative to the start of the experiment for (a, b) $CO_2$ and (c) $H_2O$ with respect to pressure for all tested materials. The first panel shows all materials together, whereas in the second and third panels, each material is plotted separately. Consistent color codes are used throughout the study.

**Figures 6: the x-axes is very disturbing, even after the highlight in the text noting that it indicates the temperature cycle, which is why the scale is not linear. There is some logic in this choice, but it discards the possibility to clearly see the temperature effect. Consider plotting with a linear temperature scale using a color code or different shapes to show the time. Another option is to use time as x-axes and superpose the temperature cycle.**

We respect our reviewers view on Figure 5 and Figure 6. However, in our opinion the plots are clearly showing both the reversibility (e.g. $H_2O$) and the irreversibility (e.g. CO and $CO_2$) of the temperature effect. Superposing a temperature cycle would pack more information on already full plots especially in the case of group 1 plots (Fig. 6a).

---

## Author Comment (AC3) · 21 Oct 2019

**Reply to the review of Anonymous Referee #1**

The authors would like to thank Anonymous Referee for the valuable comments. In the following, referee's comments are given in bold and author's responses in plain text. Suggested new text is quoted in italics together with page and line numbers.

**This paper described the testing of various materials in an aluminum chamber designed such that various materials can be inserted in the chamber and tested for adsorption effects. This paper is a follow-on from a previous paper that described the testing chamber and analytical challenges (10.5194/amt-2019-197). The paper is well-written and contains informative figures. For many of the materials tested, desorption of $CO_2$, $CH_4$, and CO were minimal, which is good news for the measurement community. However, it is not exactly clear how the results of these experiments relate to atmospheric trace gas analysis due to significant differences between this work and how compressed gas standards are used in practice.**

We understand the concern of our reviewer. The idea behind the setup and the material experiments is the following: compared to a standard cylinder (e.g. 29.5 L or 50 L) the surface to volume ratio of the small cylinders is bigger, and one can further increase the surface area by inserting materials. By testing materials at various pressure and temperature ranges, problematic materials can be identified. The following assumption lies behind these experiments: if the material has adsorbed significant amount of gas while filling the cylinder, this should be desorbed towards the end of the experiments controlled by desorption. The controlled experiments and the numerous materials tested in this study showed even under increased surface areas and lower than normally employed pressure ranges, surface effects for most materials were minimal, meaning that currently used materials are well-suited for their applications in atmospheric measurement community.

The following paragraphs will be added in discussion:

*"The presented setup enabled the investigation of surface effects under "extreme" conditions which favored adsorption/desorption. Compared to common usage in the atmospheric measurement and gas metrology communities, our study has differed in cylinder size, geometric surface to volume ratios, pressure and temperature ranges. Previous studies (Leuenberger et al. 2015, Brewer et al. 2018, Schibig et al. 2018) investigating surface effects in compressed gas cylinders have used (50 L, 10 L, or 29.5 L) cylinders. The geometric surface of the small (5 L) aluminum cylinder used in this study is 0.18 $m^2$, which results in a surface to volume ratio of 35.7 for the unloaded cylinder. Compared to 29.5 L Luxfer cylinders, the small cylinders are estimated to be more prone to adsorption by 40 %. Inserting material blocks into the aluminum cylinder further increased the surface area. Therefore, the setup allows to test materials under increased surface to volume ratios in which the surface effects should be stronger and dominant. However, despite our efforts of increasing the surface material effects were minor.*
*In addition to the properties of the materials, pressure and temperature play a role on surface effects. The following assumption lies behind the pressure experiments: if the material has adsorbed a significant amount of gas while filling the cylinder, this should be desorbed towards the end of the experiments controlled by desorption. The onset of the desorption for all tested materials except Dursan and partly DLC was observed well-below atmospheric pressures.*
*Increasing temperature is expected to facilitate desorption by providing the required energy to desorb the gas molecules from the surface and mix into gas phase. On the contrary, cooling the cylinder and its content favor adsorption and it is expected that this results in a decrease in the measured amount fraction."*

**General comments: You mention that you did not subtract the background, or blank (empty) run from the experiments. In figure 5, it would seem that the empty run shows much the same signals as the materials tested, which I suppose is why you state that only Dursan showed significant difference. I think this needs further explanation related to the other Satar et al 2019 paper (10.5194/amt-2019-197).**

We thank our reviewer for pointing this out. In order to clarify this point, the following will be added on page 9, at line 10 and the corresponding paragraph will be rearranged:

*"For $CO_2$ measurements, the amount fraction differences for all materials except Dursan were less than 0.17 µmol $mol^{-1}$, with slight differences among the various loadings. Of this difference, 0.05 µmol $mol^{-1}$ is related to the blank cylinder (background effect). The blank cylinder corresponded to the "14 bar after heating" case presented in Satar et al. (2019). More information on the blank cylinder and its filling history is provided in the above-mentioned publication. It is also crucial to consider that during all material block experiments, glass pieces were also present in the small measurement chamber. When the material runs were compared to the experiments with glass, except the DLC loading, the remaining differences were in the order of 0.02 µmol $mol^{-1}$, which corresponded to the 5 minute-standard deviation of the measured data. Moreover, during the evacuation of the measurement chamber with the DLC loading, a slightly increasing trend of -0.004 µmol $mol^{-1}$ $bar^{-1}$ was observed. For the steel loaded cylinder, the experiments where a multiport valve was upstream of the analyzer showed slightly more variation both for final amount fractions and during the pressure run. Whereas, the mass flow controller employed did not have a significant effect on the $CO_2$ measurements. "*

**On page 12, you mention that you measured the "blank" or background at the beginning at the end of the experiments, and the "blank" shown in figure 6 is the mean. How for this impact the conclusions? Do you know if the background changed smoothly over time, or abruptly as a results of adding Dursan or DLC?**

Our concern about the repeated heating cycles on page 12 line 6, was the change of the background effect of the measurement chamber only due to heating. Inserting any of the materials should not have changed the background effect of the cylinder. During the experiments the blank cylinder has evolved to be better after re-use and refilling with the same gas. Therefore, we don't think that the inserted materials can result in irreversible effects.

For clarity the sentence on page 12 line 6 is modified to:
*"This presumably resulted in a change of the background effect over the course of the presented analysis in the range of 0.04 µmol $mol^{-1}$."*

**Further, I think you can better explain why you chose the maximum deviation (page 7, line 9), given that making measurements at sub-ambient pressure introduces complications, as described in the other Satar et al 2019 paper.**

In our opinion, the maximum deviation is the most suitable parameter, since we are interested in the maximum possible effect which might occur related to the desorption process. Moreover, all data was processed using the same criterion cancelling out the effects originating from the analyzer and enable to focus on the differences between the materials.

The following will be added on page 7, line 9:
*"Maximal difference was chosen to highlight the maximum possible effect related to desorption. Since all data was processed using the same criterion, the effects originating from the analyzer is cancelled out and we focus on the differences between the materials."*

**How do these results compare with others studies (Leuenberger et al 2015; Schibig et al 2018) that suggest that a Langmuir isotherm can be used to model the physical adsorption? Your tests seem to show a much steeper increase at the lowest pressures. Is the abrupt increase at the lowest pressure related to desorption, or is it complicated by analysis at low pressure? I realize the analysis is discussed in the first Satar et al 2019 paper, but I think some important aspects need to be repeated here.**

We understand the view of our reviewer that this point needs further explanation, we have already provided more information on this comparison in the revised version of the Satar et al 2019 paper and its supplementary material. However, it is crucial to note that the aluminum cylinder without loading presented in this study corresponds to the case "14 bar aluminum after heating" and showed a maximal deviation from the initial amount fraction as low as 0.05 $\mu mol\ mol^{-1}$ making the modelling of this increase extremely difficult. For the material loadings, adsorbed amounts should be distributed between the blank cylinders and the material blocks. However as previously explained, subtracting the maximal amount fraction difference of the glass loaded cylinder from the material loaded cylinder resulted in amount fractions in the order of the standard deviation of the measured data. Therefore, for the majority of the materials fitting the Langmuir isotherm is not reasonable.

**Since it not common to use a cylinder of gas down to less than 10% of the fill pressure, what would these results say about that practice?**

These results highlight that the currently used materials are non-problematic at various temperature and pressure ranges, which is good news for the atmospheric measurements community and gas applications. The experiments were conducted under "extreme" conditions to understand possible surface effects to its full extent. Please see the explanation on the first paragraph of the replies.

**I am also concerned about how to interpret the results with respect to materials used by the scientific community. The inside of an aluminum cylinder, for example, does not typically look like the outside. Presumably the process of manufacturing a cylinder (using a ram rod) alters the surface roughness of the inside, and may deposit trace elements on the internal surface. While you tested the same aluminum alloy used for cylinders, the tested materials might not be representative of actual cylinders. Have you tested pieces of material cut from aluminum or steel cylinders? Maybe it doesn't matter since the results don't show significant desorption, but this should at least be discussed.**

We thank our reviewer for pointing this out. We have not tested materials cut from aluminum or steel cylinders. This can be done within the scope of another study. Regarding other tested materials, commercial coatings aiming to provide an inert surface undergo specific cleaning and chemical or physical vapor deposition of processes, which should be consistent regardless of the coated piece. Concerning copper and brass, these materials might be used in regulators or as seals, and not commonly used for cylinders. Unfortunately, one individual study with a new setup is not capable of answering of all questions about materials employed in the atmospheric measurement community. The study was a first step in understanding surface effects by increasing surface areas, for a significant number of materials under various pressure and temperature ranges with a sufficient number of replicates.

The following will be added into discussion:

*"Testing pieces cut from the aluminum and steel cylinders commonly used in the community would be a valuable addition to enable direct comparison between the commonly used cylinder materials and the produced material blocks at low pressures and high temperatures. "*

**Specific Comments**

**P1, L5: suggest: "For this study we used small-volume chambers designed to be used for adsorption studies". –** text modified accordingly

**P3, L30: suggest: "A mother cylinder was...." –** text modified accordingly

**Figure 4: Minor point: I'm not sure of the significance of the box plots based on only 3 data points. I guess you are assuming normal distributions and assigning quartiles based on the standard deviation.**

We thank our reviewer for this comment. In the box plots R uses the "fivenum" function which is the descriptive statistics for minimum, $1^{st}$ quartile, median, $3^{rd}$ quartile and the maximum. In case of 3 data points, the $1^{st}$ quartile is the mean of the minimum value and the median, whereas the $3^{rd}$ quartile is the mean of the median and the maximum value. Therefore, the whiskers extend to the minimum and the maximum values, respectively. Since in Fig. 4 and Fig. 7, all data points are shown in addition to the box plots, using box plots for the figures is non-problematic.

On page 7, at line 10, we will add the following:

*"The median is denoted by the horizontal line, whereas the mean is shown by the square. Since for most cases only 3 replicates are present, the $1^{st}$ quartile is the mean of the minimum and the median, whereas the $3^{rd}$ quartile is the mean of the median and the maximum value. For clarity, data points used for the box plots are also shown and they are denoted by the black points. "*

We have also rearranged Fig.4 to better distinguish the mean (denoted by the squares) from the results of the replicates (denoted by the black points)

[Figure]

**Figure 4.** Box plots for all materials for the species (a) $CO_2$, (b) zoom-in for $CO_2$, (c) CO, (d) $CH_4$ and (e) $H_2O$. y-axes show the maximal amount fraction difference relative to the initial amount fraction. Horizontal lines in each box plot shows the median, whereas the square in the center of the box is the mean of the maximal amount fractions of the replicates.